# Validation of a Mathematical Model Describing the Dynamics of Chemotherapy for Chronic Lymphocytic Leukemia In Vivo

**DOI:** 10.3390/cells11152325

**Published:** 2022-07-28

**Authors:** Ekaterina Guzev, Suchita Suryakant Jadhav, Eleonora Ela Hezkiy, Michael Y. Sherman, Michael A. Firer, Svetlana Bunimovich-Mendrazitsky

**Affiliations:** 1Department of Mathematics, Ariel University, Ariel 4070000, Israel; ekaterin.shevcove@msmail.ariel.ac.il; 2Department of Chemical Engineering, Ariel University, Ariel 4070000, Israel; jadhavs@ariel.ac.il (S.S.J.); firer@ariel.ac.il (M.A.F.); 3Department of Molecular Biology, Ariel University, Ariel 407000, Israel; elahizkia@gmail.com (E.E.H.); sherma1@ariel.ac.il (M.Y.S.); 4Adelson School of Medicine, Ariel University, Ariel 4070000, Israel; 5Ariel Center for Applied Cancer Research, Ariel University, Ariel 4070000, Israel

**Keywords:** A20 cells, cytotoxicity rate, in vivo experiments, logistic cancer growth rate, mathematical model, personalized chemotherapy, stability analysis, tumor dynamic

## Abstract

In recent years, mathematical models have developed into an important tool for cancer research, combining quantitative analysis and natural processes. We have focused on Chronic Lymphocytic Leukemia (CLL), since it is one of the most common adult leukemias, which remains incurable. As the first step toward the mathematical prediction of in vivo drug efficacy, we first found that logistic growth best described the proliferation of fluorescently labeled murine A20 leukemic cells injected in immunocompetent Balb/c mice. Then, we tested the cytotoxic efficacy of Ibrutinib (Ibr) and Cytarabine (Cyt) in A20-bearing mice. The results afforded calculation of the killing rate of the A20 cells as a function of therapy. The experimental data were compared with the simulation model to validate the latter’s applicability. On the basis of these results, we developed a new ordinary differential equations (ODEs) model and provided its sensitivity and stability analysis. There was excellent accordance between numerical simulations of the model and results from in vivo experiments. We found that simulations of our model could predict that the combination of Cyt and Ibr would lead to approximately 95% killing of A20 cells. In its current format, the model can be used as a tool for mathematical prediction of in vivo drug efficacy, and could form the basis of software for prediction of personalized chemotherapy.

## 1. Introduction

Recent advances in mathematical modeling have led to the development of ordinary differential equation-based (ODE) models capable of predicting the outcome of complex biological systems, including tumor growth [1,2]. These models have led to interesting applications regarding tumor growth biology [3,4], and predictions regarding the selection of effective therapeutic protocols aimed at reducing the toxic side effects of chemotherapy [5,6,7,8,9,10]. However, most of these studies have focused on solid tumors, while few computational models have been used to study blood cancers [11,12,13,14]. As parameters such as cell growth and survival dynamics and cellular interactions within the tumor microenvironment likely differ considerably from solid tumors to leukemias and lymphomas, it is expected that several parameters of ODE computational models need to be adapted for blood-borne cancers [15].

Recently, several studies reported mathematical simulation models describing the dynamics of blood cancers under the influence of various small molecule chemotherapeutic drugs and/or immunotherapy [16,17,18,19]. These studies showed that quantitative analysis could be employed to investigate whether, for example, the duration of therapy could be reduced without increasing the risk of relapse [16], or to compare the efficacy of different treatment protocols [17].

The current models have several important limitations. First, they are built upon simulation data rather than real-life experiments. Second, it is important to develop models that can be applied for personalized chemotherapy dosing by taking into account the variability of tumor cell growth rate between patients [20,21]. The ability to personalize drug dosages should also lead to a reduction in side effects from off-target drug cytotoxicity [22]. Third, many studies have demonstrated conflicting results between optimal drug doses determined by in vitro versus in vivo experiments [23,24]. This disparity highlights the difficulty in translating drug efficacy data from artificial cell culture conditions to the complexity of the whole body.

In the present study, we aimed to address the first two issues noted above in relation to the blood cancer Chronic Lymphocytic Leukemia (CLL). CLL is a malignancy of B lymphocytes that accumulate especially in the blood, but also in lymph nodes and spleen. It is one of the most common adult blood cancers in the western world [25]. Over the last decade, the introduction to the clinic of new, molecular targeted, small-molecule drugs such as the Bruton tyrosine kinase (BTK) inhibitor Ibrutinib (Ibr) and its newer derivatives has transformed CLL therapy and contributed to the extended overall survival of patients [26]. Nonetheless, CLL patients can develop drug resistance and suffer from toxic side effects, and the disease remains incurable [27]. Therefore, further improvements in mathematical models are needed to assist clinicians with designing more effective treatment protocols.

To build our model, we first determined the growth rate of murine A20 leukemic cells in immuno-competent Balb/c mice. This allowed us to formulate the logistic dynamic of these cancer cells. Then, we conducted experiments in vivo to compare the cytotoxicity of two drugs—Cytarabine (Cyt), an inhibitor of the enzyme topoisomerase [28] and Ibr against the leukemic cells. The drug doses were selected by reviewing the literature on the use of these drugs in in vivo models of cancer [29,30,31,32,33,34]. The results afforded calculation of the killing rate of the A20 cells as a function of therapy. We compared our experimental data with the simulation model and validated the latter’s applicability. A simplified work scheme below (Figure 1) illustrates the whole process.

Since our model is based on in vivo experiments it is more relevant to real-life situations. By calculating the growth rate of the cancer cells from each individual, we proposed that the model can simulate the optimal dose of the chemotherapeutic drug, a critical step towards achieving personalized chemotherapy.

## 2. Materials and Methods

### 2.1. Cells and Drugs

A20 murine leukemic cells, originally obtained from the ATCC, were transfected with a monomeric red fluorescent protein (mCherry) and grown in RPMI 1640 medium supplemented with 10% Fetal Bovine Serum (FBS) (Thermo Fischer, Waltham, MA, USA), 1% L-Glutamine and 0.33% Pen-Strep solution. Cells were maintained at 37 °C and 5% CO_2_.

### 2.2. In Vivo Experiments

Animals were purchased from Envigo (Jerusalem, Israel). The in vivo experiments were approved by the Ariel University Animal Ethics Committee (Permit number IL-216-01-21). Mice (BALB/c, female, 6 weeks, 17–19 g weight) were maintained in a controlled environment (22 °C) with free access to food and water.

To measure the growth rate of mCherry A20 cells in vivo, 20 mice were inoculated via the tail vein with $5\times 10^{4}$ logarithmic phase cells in Phosphate-Buffered Saline (PBS). Based on previous preliminary calibration experiments, we knew that A20 cells begin to proliferate and appear in blood between two and three weeks following inoculation (data not shown). Therefore, on Day 16 after inoculation, blood was collected from the tail veins of four randomly chosen mice, and again on Day 22 from four other mice. The samples were treated with Red Blood Cell lysis buffer (ebioscience) for 10 min at room temperature. The reaction was stopped by the addition of an equal volume of PBS, followed by centrifugation at 300× *g* for 10 mins at room temperature. The cell pellet was resuspended in FACS buffer, and A20mCherry positive cells were measured on a CytoFLEX (Beckman Coulter) flow cytometer. Acquired data were analyzed using FlowJo software.

On Day 22 after cell inoculation and prior to initiation of drug therapy, blood was taken from the tail vein of each mouse, and mCherry A20 cells were measured as described above. Drug treatment protocols using Cyt and Ibr were derived after comparing several published protocols in which these drugs had been used in in vivo studies (Cyt- [29,30,31]; Ibr- [32,33,34]). The 20 inoculated mice were randomly divided into 5 groups: Group 1: the Control group, which only received vehicle (PBS); Group 2: the Cyt Low group, which received 0.12 mg/kg of Cyt for 5 consecutive days; Group 3: the Cyt High group, which received 62.5 mg/kg of Cyt for 3 consecutive days; Group 4: the Ibr Low group, and Group 5: the Ibr High group, which received 9mg/kg and 18 mg/kg of Ibr, respectively, on days 1–5 and 8–10 from the beginning of the treatment. Blood was collected from the tail vein from all mice on Day 12 after the beginning of treatment. The frequency of A20 mCherry cells in the blood samples was measured using flow cytometry, as described above. The difference in the frequency of these cells in the blood before and after treatment was used to calculate the leukemia growth index of each group of mice. The difference in growth index between treated and non-treated mice was used to calculate the inhibition of cell growth as a function of treatment.

### 2.3. Validation and Numerical Simulations of the Model

The data from the in vivo experiments and the parameters from Table 1 were used to validate our model. Computer simulations were performed using fourth-order adaptive step Runge–Kutta integration, as implemented in the ODE45 subroutine of MATLAB [35].

## 3. Results

### 3.1. Determination of A20 mCherry Growth Rate

In order to determine the growth rate in vivo of A20 mCherry cells, tail vein blood was taken on Day 16 and Day 22 after inoculation of cells, each time from four different mice (Figure 2). The average number of A20 mCherry cells in the first and second bleeds was assessed by positively gating cells using FSC and SSC parameters; doublets and debris were eliminated from the analysis. The data were used to calculate the growth rate according to the formula:$$r=\frac{\ln\left(N(t)/N(0)\right)}{t},$$
where
N(t) = the number of cells at time *t*;N(0) = the number of cells at time 0;*r* = growth rate;*t* = time (usually in hours).

Thus, according to Figure 2, when N(0)=3662; N(t)=16,338 and t=144 h, the growth rate of A20 mCherry cells will be:$$r=\frac{\ln\left(16,338/3662\right)}{144}=0.01\;\left(h^{-1}\right).$$

### 3.2. Drug Cytotoxicity In Vivo

To determine the efficacy of Cyt and Ibr in vivo, five groups of mice were treated with different doses of drugs and time periods, as described above. Tail vein blood was collected twice from each mouse, on the day of initiation of treatment and two days after the last treatment. At each time point, the percent change in frequency of A20 cells in each treated mouse relative to the average frequency in the Control (untreated) group was calculated. From these data, the average A20 frequency change for each treated group was obtained. The difference between the average frequency in each test group and the untreated Control group represents the percent growth inhibition as a function of treatment. The results (Figure 3) show that the growth inhibition due to Cyt treatment was dose dependent (low dose 9%, high dose 58%), whereas inhibition of growth due to Ibr was not (low and high doses both about 44 %). The results are summarized in Table 2.

### 3.3. Formulation of the Model

Based on previous studies [36,38] and our in vivo experiments described above, we formulated an ODE model to mathematically explain the interaction between CLL cells and chemotherapeutic drugs in vivo:$$\left\{\begin{array}{cc} \frac{dA}{dt}=rA\left(1-\frac{A}{K}\right)-\mu_{A}AE-\frac{\mu_{AC}AC}{a+C}, & \left(1\right)\\ \frac{dE}{dt}=-\mu_{E}E+\frac{pAE}{c+A}-\mu_{EA}AE-\frac{\mu_{EC}EC}{b+C}, & \left(2\right)\\ \frac{dC}{dt}=\sum_{m=0}^{N-1}d\delta \left(t-m\tau \right)-\mu_{C}C-\frac{\mu_{CA}CA}{a+A}, & \left(3\right) \end{array}\right.$$

$\frac{dA}{dt}$ describes the dynamic of A20 mCherry cells. It is comprised of three terms. One is positive, corresponding to the logistic cancer growth characterized by the coefficient, *r*, which is limited by the maximal tumor cell number, *K*. The second term is negative, corresponding to living cells becoming dead due to the interaction with effector cells with the rate $\mu_{A}$. The last negative term represents the log-kill hypothesis [36], with a Michaelis–Menten drug saturation response [39], a+C; $\mu_{AC}$ is the death rate resulting from the action of the drug on cancer cells.

$\frac{dE}{dt}$ describes the dynamic of immune effector cells. $\mu_{E}$ describes the natural death rate of effector cells; *p* describes the production rate of effector cells stimulated by the cancer cells; *c* describes the number of cancer cells by which the immune system response is half of its maximum; $\mu_{EA}$ describes the interaction coefficients between cancer and effector cells affecting immune populations; $\mu_{EC}$ describes the mortality rates due to the action of the chemotherapeutic drug on effector cells; *b* describes the drug amount for which such effects are half of its maximum in the immune cell population.

$\frac{dC}{dt}$ describes the first-order pharmacokinetics of a drug [40] with an external source. A dose, *d*, of the chemotherapeutic drug is injected every τ hours.

By modeling the injection as a shifted Dirac Delta function δ(t−mτ), m=0,1,…,N−1, the *m*-th dose raises C(t) by exactly *d* units at t=mτ, m=0,1,…,N−1. A full explanation of the Dirac Delta function in its continuous-time version is given in Appendix B.

$\mu_{C}$ is the deactivation rate calculated by the formula $\mu_{C}=\frac{ln(2)}{t_{1/2}}$, where $t_{1/2}$ is the elimination half-life between 1–3 h (biphasic) for Cyt and 4–6 h for Ibr (www.drugbank.ca).

$\mu_{CA}$ is the rate at which drug molecules attack cancer cells. The parameter *a* represents the drug concentration which produces 50% of the maximum activity of the drug in the A20 mCherry cell population [41].

We performed a mathematical analysis of our model (1)–(3) by identifying fixed points and their stability, and found that the system is characterized by three fixed points, one of which is asymptotically stable (Appendix A).

### 3.4. Estimation of the Parameters of the Model (1)–(3)

In this section, we evaluate model parameters (Table 1) together with the detailed methods and the literature sources for their evaluation. To have biological meaning, all values of the parameters must be positive. We specify the initial conditions at t = 0 as:$A\left(0\right)=5\times 10^{4}$ (cells/mouse)—the initial number of A20 cells;E(0)=2500 (cells/mouse)—the initial number of effector cells;C(0)=0, before the treatment.

The number of drug molecules was calculated using the expression: $\frac{m\times Na}{M},$ where
*m* = the mass of drug in kg;Na = avogadro number = $6.022\times 10^{23}$ (constant);*M* = the molar mass of drug (Cyt 243.217 g/mol; Ibr 440.5 g/mol.).

For example, for 1.25 mg/mouse (62.5 mg/kg) of Cyt, i.e., 1.25 mg = 0.00000125 kg, the number of Cyt molecules will be:$$\frac{0.00000125\times 6.022\times 10^{23}}{243.217}=3\times 10^{15}=1.25\times 2.4\times 10^{15};$$
for 0.36 mg/mouse (18 mg/kg) of Ibr, the number of drug molecules will be:$$\frac{0.00000036\times 6.022\times 10^{23}}{440.5}=5\times 10^{14}=0.36\times 1.4\times 10^{15};$$

### 3.5. Sensitivity Analysis of Parameter $\mu_{AC}$

A sensitivity analysis was conducted to determine the range of parameter $\mu_{AC}$, which has the most significant impact on the cancer cell population under the influence of different drugs. We aspired to fit the parameter $\mu_{AC}$ as accurately as possible for each treatment protocol in order to achieve the same growth inhibition as in the real-life experiments on mice. As shown in Figure 3 we built a mini model for each drug concentration (Figure 4A–D). The most appropriate value of $\mu_{AC}$ was obtained with curve (3), highlighted in red on each sub-graph.

Figure 4A, the curve (3) shows the impact on A20 mCherry cells under influence of a low dose—0.12 mg/kg of Cyt that was injected for five days. The growth inhibition at the end of this period was only about 10%. Under influence of a high dose of Cyt—62.5 mg/kg, growth inhibition was about 59% (Figure 4B, the curve (3)). In Figure 4C and in Figure 4D, the curves (3) show around 44% growth inhibition with both 9 mg/kg and 18 mg/kg of Ibr. Figure 4 also simulates the comparative growth rates of untreated versus treated A20 cells for each treatment for the whole mouse. In contrast, Figure 3 represents the inhibition of A20 cell growth, as measured in 200,000 cells in 50 uL of blood. Although it is estimated that the whole-body blood volume of Balb/c mice is approximately 2.5 mL, we did not consider this accurate enough for our purposes. Therefore, we could not compare the simulation and experimental data for the actual number of A20 cells in the whole animal in the one graph. Instead, to validate the model, we compared the % inhibition of growth for each therapeutic protocol obtained experimentally and by simulation, which illustrates excellent accordance between the two results (Figure 5).

### 3.6. Prediction of the Effect from Combined Therapy

To test the growth inhibitory effects of Cyt combined with Ibr, we ran a simulation using concentrations from Table 2, 62.5 mg/kg of Cyt and 9 mg/kg of Ibr for 8 days of treatment (5 days treatment, 2 days break, 3 days treatment) and adjusted the calculated parameters into the model (1)–(3): $C\left(0\right)=3.25\times 10^{15}$ drug molecules. Under these conditions, Cyt composes about 92% of the molecules, and Ibr about 8%. Hence, $\mu_{C}$ will be: (0.231×0.92)+(0.116×0.08)=0.221; $\mu_{AC}$ will be 0.012+0.0041=0.0161. This simulation (Figure 6) predicted 95% of cell growth inhibition, which represents an 85% increase in killing efficiency compared to separate treatment.

## 4. Discussion and Conclusions

The clinical outcome for patients with CLL has significantly improved over the last decade, although the disease remains incurable and new treatment approaches are clearly needed. In this study, we present an animal-based ODE model (1)–(3) that describes the dynamic of CLL cell growth in vivo under the influence of different cytotoxic drugs. We did not measure immune cells in this study, but rather used parameters employed by others in the literature to describe the interaction of immune cells with the cancer cells. We applied a multistep approach to validate the model (1)–(3), using experimental in vivo data with different concentrations of either Cyt or Ibr. Our numerical simulation results indicate that the model predicts the response of leukemic cells in mice to chemotherapy (Table 2).

In a recent study, we found that Cyt—a drug not currently used in CLL therapy—was more cytotoxic to A20 cells in vitro than Ibr [38,42], which suggested that the model could be used to predict the repurposing of cancer drugs, a subject which is attracting much attention [43]. Indeed, this prediction was vindicated when tested in our animal model (Figure 3).

To demonstrate an additional application of our model, we performed a numerical simulation (Figure 6) of the potential effect of Cyt plus Ibr on A20 leukemic cells, which predicted that such a combination could significantly increase cytotoxicity and inhibit cancer cell growth by up to 95%. It would now be valuable to test this combined treatment in vivo, especially as these drugs have different modes of action.

Our model exhibits several stable states that depend on biologically related parameters. Three equilibria were investigated (Appendix A). We found that $Eqm_{0}^{*}$ and $Eqm_{1}^{*}$ equilibria occur when there is no treatment. Analysis of the stability of the system shows that the free-tumor $Eqm_{0}^{*}$ equilibrium is not stable. This means that if there are no more cancer cells and the treatment is stopped, the model is in equilibrium without growth, albeit unstable. This may also represent a state of cancer cell dormancy, an adaptive strategy used by cancer cells to overcome drug cytotoxicity [44]. This stage may persist until complete metabolism of the drug, which would allow the cell to re-enter the cell cycle and tumor growth to recur. The fixed point $Eqm_{1}^{*}$ is a stable equilibrium, and the system reaches this equilibrium when the number of cancer cells reaches its maximum in the body. The system is not stable at $Eqm_{2}^{*}$ equilibrium with periodical chemotherapy. We obtain $Eqm_{2}^{*}$ when treatment is stopped and before the cancer cells are completely removed, and then the system tends to its unstable equilibrium.

In this study, we aimed to address two important limitations in current mathematical models describing chemotherapy effects in blood cancers such as CLL. These models are not built on actual in vivo experiments. Therefore, we first calculated the growth rate of the A20 cancer cells in mice, which lead to the logistic dynamic of these cancer cells. This step is necessary to carry out for each type of cancer cell, and will be easier to perform for blood-borne cancers such as leukemia than for solid tumors.

Current models also do not easily lend themselves to personalized chemotherapy dosing, partially because tumor cell growth rates vary between patients. By validating our experimental model with simulations studies (Figure 5), we were able to select an optimal range of drug dosages to test. Furthermore, the model can be used to simulate combination drug therapy. We found that in this way, we could predict that the combination of Cyt and Ibr would lead to approximately 95% killing of A20 cells (Table 2). Such high rates of killing are not expected in clinical practice, mainly due to subsequent toxicities. The response rates and toxicity profiles induced by Cyt therapy vary across patients, mainly due to variation in cancer types, treatment protocols, and genetic polymorphism in cytarabine-metabolizing enzymes [45,46]. Nonetheless, Cyt is still commonly used in the chemotherapy of a number of cancer types. The goal of our study was not to demonstrate the utility of Cyt for the treatment of CLL. Rather, our model allowed us to predict a potentially effective new combination of two drugs. Further calibration experiments in vivo may reveal that as the two drugs we tested have different modes of action, in practice, it may be possible to further reduce the dosages and still obtain acceptable efficacy. These doses will, as stated above, also depend on the actual growth rate of a patient’s cancer cells.

It remains difficult to correlate mathematical models of drug efficacy based on in vitro experiments’ in vivo outcomes. We have developed ODE models for drug efficacy in leukemia derived from in vitro cytotoxicity experiments [38,42] that are able to predict and compare the cytotoxic efficacy of traditional CLL drugs, such as Chlorambucil and Melphalan, with that of Ibr and Cyt. However, the predicted optimal doses obtained from these models did not correlate with those derived from the in vivo models described here. A possible solution to this problem could be to use methods of Artificial Intelligence to derive appropriate algorithms which can bridge between the different parameters of in vitro and in vivo environments.

The ODE model presented in this study (1)–(3) can be adapted to various types of cancer cells and different chemotherapeutic drug doses, so long as the growth rate of the cells and the cytotoxic efficacy of the drugs are known. We believe that experimentally validated mathematical models such as these will be critical for development of tools to more accurately predict therapeutic efficacy of drug therapy for cancer.

## Figures and Tables

**Figure 1 cells-11-02325-f001:**
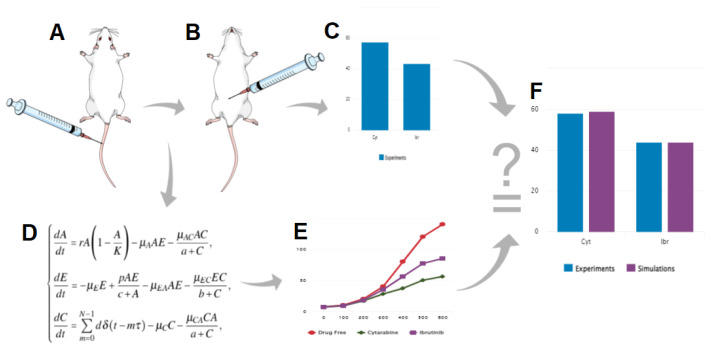
Schematic representation of the workflow used in this study. (**A**), injection of A20 leukemic cells; (**B**), drug treatment; (**C**), in vivo examination of the effect of a drug on cancer cell viability; (**D**), development of the mathematical model; (**E**), numerical simulations of the model; F, validation of the model by correlation of experimental results and simulation.

**Figure 2 cells-11-02325-f002:**
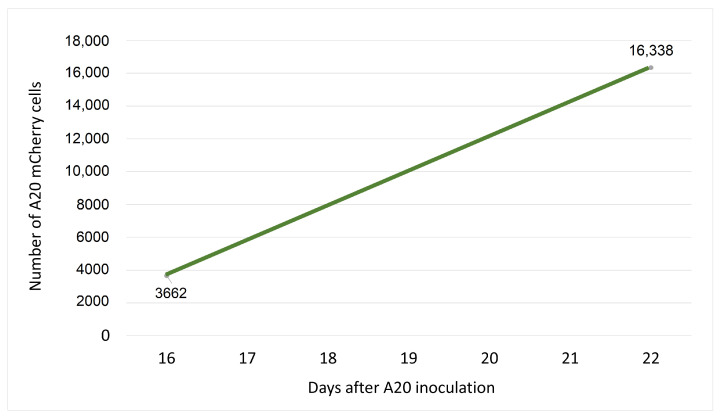
Growth of A20 cells in vivo. Twenty Balb/c mice were inoculated via the tail vein with $5\times 10^{4}$ A20 cells. On Day 16, blood was collected from the tail vein of four randomly chosen mice, and again on Day 22 from four other mice. A20 mCherry positive cells were measured on a CytoFLEX (Beckman Coulter) flow cytometer. Acquired data were analyzed using FlowJo software. The points on the graph represent the average number of A20 mCherry cells measured among $2\times 10^{5}$ blood cells analyzed from 4 mice. These data were used to calculate the growth rate of the cells in vivo.

**Figure 3 cells-11-02325-f003:**
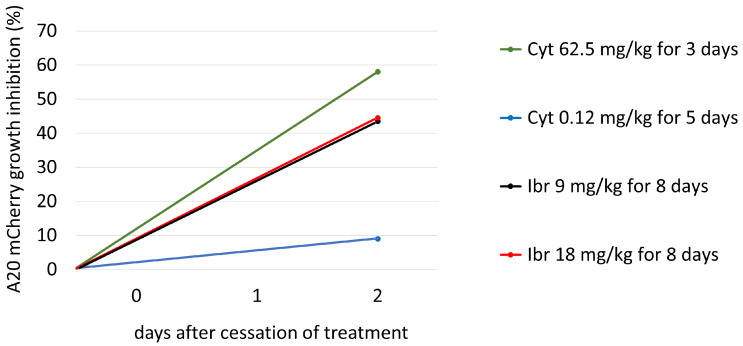
The growth inhibition of A20 mCherry cells in vivo under the influence of different treatment protocols. On day 22th, after A20 inoculation, blood was taken from the tail vein of each mouse, and A20 cells were measured as described above. Cyt Low group (blue bar) received 0.12 mg/kg of Cyt for 5 consecutive days; Cyt High group (green bar) received 62.5 mg/kg of Cyt for 3 consecutive days; Ibr Low (black bar) and Ibr High (red bar) groups received 9mg/kg and 18 mg/kg of Ibr, respectively, on days 1–5 and 8–10 from the beginning of the treatment. Blood was collected from the tail vein on Day 12 after the beginning of treatment from all mice. The frequency of A20 was measured using flow cytometry, as described above. The difference between the average frequency in each test group and the untreated Control group equals percent growth inhibition as a function of treatment.

**Figure 4 cells-11-02325-f004:**
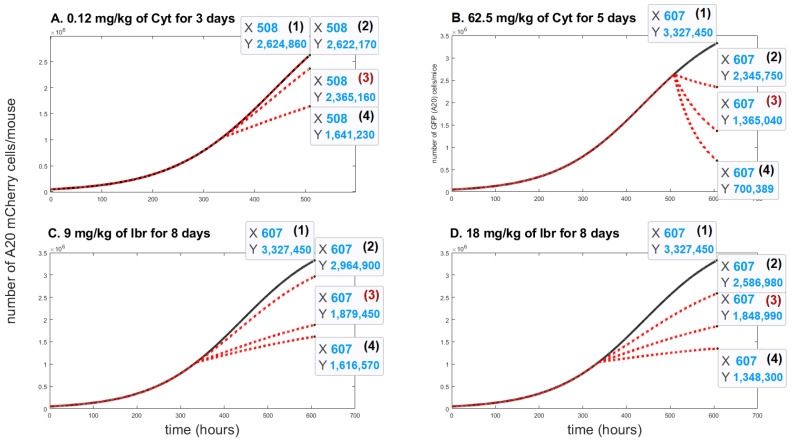
The time evolution of A20 mCherry cells without any treatment (**black** solid lines) and with different values of the death rate resulting from the action of the drug, $\mu_{AC}$ (**red** doted lines). Three values were considered for each treatment protocol: In Figure 4A (2) $\mu_{AC}=0.0001$; (3) $\mu_{AC}=0.001$; (4) $\mu_{AC}=0.003$. In Figure 4B, (2) $\mu_{AC}=0.005$; (3) $\mu_{AC}=0.012$; (4) $\mu_{AC}=0.02$. In Figure 4C, (2) $\mu_{AC}=0.001$; (3) $\mu_{AC}=0.0041$; (4) $\mu_{AC}=0.005$; In Figure 4D, (2) $\mu_{AC}=0.002$; (3) $\mu_{AC}=0.0042$; (4) $\mu_{AC}=0.006$.

**Figure 5 cells-11-02325-f005:**
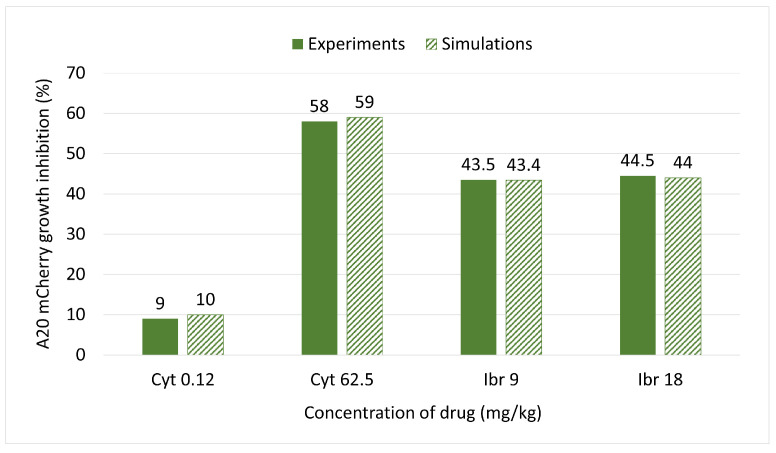
Comparison of inhibition of A20 mCherry cell growth under the influence of different doses of Ibr or Cyt between the model simulations (textured bars) and experimental data (fill bars).

**Figure 6 cells-11-02325-f006:**
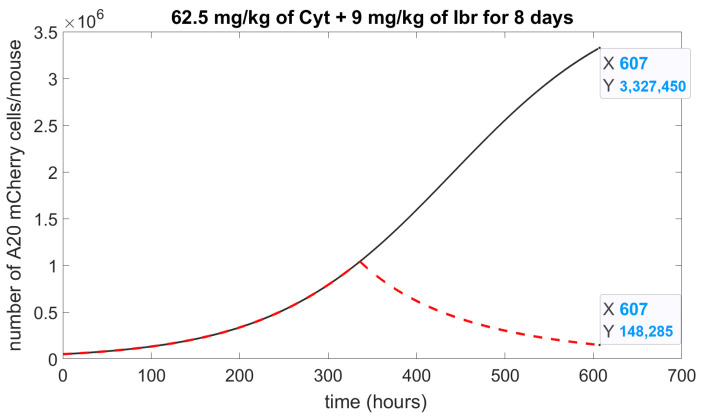
Numerical simulation of drug combination. The simulation of the model (1)–(3) represents the number of A20 mCherry cells affected by the effect of Ibr with Cyt. A solid **black** curve is a control, without a drug; a dashed **red** curve is 62.5 mg/kg of Cyt and 9 mg/kg of Ibr; Initial concentration of A20 mCherry cells, $A\left(0\right)=5\times 10^{4}$.

**Table 1 cells-11-02325-t001:** Table of parameters based on experimental results related to the model of Equations (1)–(3).

Parameter	Physical Interpretation (Units)	Estimated Value	Reference
*r*	A20 growth rate [h^−1^]	0.01	Experimental data
*K*	Maximal tumor cell population [cells/mouse]	$4\times 10^{6}$	Simulation
$\mu_{AC}$	Cytotoxicity rate in the presence of drug [h−1]	0.001—Cyt 0.12 mg/kg 0.012—Cyt 62.5 mg/kg 0.0041—Ibr 9 mg/kg 0.0042—Ibr 18 mg/kg 0.0161—Cyt 62.5 + Ibr 9	Simulation
$\mu_{CA}$	Deactivation rate of drug due to killing of A2 cells [h^−1^]	$\mu_{AC}\times 10$	Simulation
*d*	Periodic administration dose [molecules/mouse]	Cyt—dose (mg/mouse) $\times 2.4\times 10^{15}$ Ibr—dose (mg/mouse) $\times 1.4\times 10^{15}$	Experimental data
$\mu_{C}$	Chemical deactivation rate of drug [h^−1^]	Cyt—0.231 Ibr—0.116	[36]
$\mu_{A}$	Interaction coefficients between effector and A20 cells [h^−1^]	$2\times 10^{-12}$	-
*a*	Drug amount that produces 50% maximum effect in A20 cell population [molecules]	$2\times 10^{3}$	-
*p*	Production rate of effector cells stimulated by A20 cells [h^−1^]	$4\times 10^{-14}$	-
$\mu_{EC}$	Mortalities rate due to the action of the chemotherapeutic drug on effector cells [h^−1^]	417	-
*c*	Number of A20 cells by which the immune system response is the half of its maximum [cell]	$10^{2}$	-
*b*	Drug amount that produces 50% maximum effect in immune cell population [molecules]	$5\times 10^{6}$	-
$\mu_{EA}$	Interaction coefficient between A20 and effector cells [h^−1^]	$4\times 10^{-15}$	[37]
$\mu_{E}$	Natural death rate of effector cells [h−1]	$4\times 10^{-5}$	-

**Table 2 cells-11-02325-t002:** Results of Experiments and Simulations.

Concentration of Drugs (mg/kg)	Cell Growth Inhibition (%) from the Simulation Data	Cell Growth Inhibition (%) from the Experiment Data
Cyt 0.12	10	9
Cyt 62.5	59	58
Ibr 9	43.4	43.5
Ibr 18	44	44.5
**Cyt 62.5 + Ibr 9**	**95**	-

## Data Availability

Data are available upon written request from the authors.

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
