# Peer review of "Validation of a Mathematical Model Describing the Dynamics of Chemotherapy for Chronic Lymphocytic Leukemia In Vivo"

_cells, 2022, doi:10.3390/cells11152325_

Round 1
Reviewer 1 Report
Interaction between immune cells and other cells has more complicated form.
Total quality of chimeotherapeutic must be restricted.
For viable therapy it is necessary to add restriction on total number of cancer and immune cells. Thus this problem is optimal control problem with phase constrains.
Author Response
Answers to reviewers
We would like to thank the reviewer for their comments and suggestions which have led to an improved manuscript, careful review, and for making sure we produce the best academic manuscript we can. We altered the Introduction and Conclusion sections according to the reviewers’ suggestions and completely changed the Abstract. Specifically, we introduce some corrections to Figures 1, 3, 4, add 5. In addition, we added a section (3.5) that outlines in detail the experiment design and model fitting processes. Please note, that all the changes we made can be found in the text highlighted in yellow.
Reviewer 1:
- Interaction between immune cells and other cells has more complicated form.
Answer: We agree with the reviewer regarding the interactions between immune cells and cancer cells. However, the objective of the proposed manuscript is to model the cytotoxic effect of the Cytarabine and Ibrutinib drugs in A20-bearing mice. As a result, we did not measure anything about the immune cells. As an alternative, we used parameters employed by others in the literature to describe the interaction between immune cells and cancer cells (Rodrigues et al. 2019, Kuznetsov et al. 1994, de Pillis et el. 2006). Following this comment, we introduced a detailed explanation of this manner to the Discussion Section, Pages 10-11.
- Total quality of chimeotherapeutic must be restricted.
Answer: Thank you for pointing this shortcoming out. The dose range of each drug was derived from previously published studies in which the authors used similar models [15-20]. In preliminary experiments (not shown) we calibrated the dosage of drugs used and found their lethal dose. The doses used in the experiments were below these lethal doses. For example, 100 mg/kg of Cytarabine was lethal, therefore we chose to test 62.5 mg/kg and 0.12 mg/kg.
- For viable therapy it is necessary to add restriction on total number of cancer and immune cells. Thus this problem is optimal control problem with phase constrains.
Answer: Thank you for this suggestion. In the present study our aim was to develop a model based on the experimental data and then validate the In silico results using the obtained in vivo results. In this way, we show one is able to predict the optimal dose of combined therapy using the proposed model and simulation, as shown in Section 3.7. Nonetheless, following this comment, we plan in an upcoming study to add an optimal control function to the model
Reviewer 2 Report
In their paper “Validation of a Mathematical Model Describing the Dynamics of Chemotherapy for Chronic Lymphocytic Leukemia In vivo” the authors describe a mathematical model for describing the experimentally observed impact of ibrutinib and cytarabine administered to a murine CLL model using different dosing protocols. The authors’ mathematical model captures the final tumor growth inhibition (TGI) for the different dosing regimens and is used by the authors to simulate the projected impact of combination treatment.
My chief concern with this work is that at the current state, it is quite superficial. Below I will attempt to provide suggestions that may improve the work after significant revisions.
1) The question that the authors are trying to address is not very clear – “write a model and fit it o data and run simulations” is the “how” of the work, not the “why”. For that, the introduction needs to be worked significantly to clarify the question that can be addressed with these methods (“cure cancer” is not it), gaps in knowledge, how the work fills these gaps, what we now know that we didn’t know before this work, and what the next steps are. These points are not clear.
2) Diagram in Figure 1 seems out of order – as of right now it looks like A, B and C are done separately, and D and E are done separately, and then they converge to F. My understanding of the flow is that A->B->C->D->E->F, and if that is the case, the arrows need to be redrawn.
3) Figure 3: I’m assuming it is reporting the final TGI at the time of when the animals were sacrificed. This is not clear from this diagram. Please redraw it as longitudinal graph, with time on the x axis, and tumor volumes under different regimens on the y-axis. If the graph gets cluttered because of multiple data points, use different subplots.
4) Model: the terms are not well justified.
a. I take issue with using a mass action term to describe tumor-immune interactions, since it has the obvious flaw of predicting high tumor cell kill even when population sizes of either immune or cancer cells are very small. This is obviously incorrect. I would suggest using a ratio-dependent term, such as described in classic works by DePillis and Radunskaya (xy/(x+y), with appropriate parameters), as it corrects this issue.
b. The tumor-dependent growth and death terms in the equation for effector cells are not justified – what are the mechanisms this is meant to capture? If you have references, please cite them. If this is purely phenomenological, that’s ok (albeit weaker of course), but please state that clearly.
c. Do we know that both these drugs can cause immune cell mortality to account for the last term in dE/dt? If so, please provide references.
d. Why is the drug cleared through interactions with cancer cells? Please provide explanation and references.
5) Figure 4. Please plot data on the same graphs as the simulation results. A claim that there is “excellent accordance” between data and results is currently unsupported.
6) Drug synergy is a very specific term, typically defined through the Blys criteria. I am assuming this was not the criterion used here? If so, please change the wording as it currently is not correct to describe the projected impact of the combination.
7) Mathematical analysis in the appendix is nice but not very helpful – stability analysis is not very informative at this stage. Ok, solution exists, and we know its stability. What does that tell us about the biology? What are the important parameters responsible for where the equilibrium will be? What insights were gained from the stability analysis, or is it just a mathematical formality?
a. More informative would be sensitivity analysis (which parameters and corresponding mechanisms are particularly sensitive to changes in system values, to changes in treatment)? Identifiability analysis would be more informative as well – how do you know these parameter combinations are unique?
8) Parameter estimates: how were they obtained? Methods are missing. Did you use fmincon? Monte Carlo Simulations? Found them by hand?
9) There is no analysis of what may be driving “synergy”, or additivity, or anything else. What terms are responsible? What are the possible biological interpretations?
10) Paper is severely undercited, and primarily cites mathematical papers, which would perhaps be appropriate (with the addition of sensitivity and potentially also identifiability analysis) in a mathematical-leaning journal but not in Cells. This model is potentially a good start but it is currently very far from being useful for biological purposes to even be listed in “methods”.
11) Technical aspect: typically the easiest way to implement modeling drug administration is through change in initial conditions for the equation for drug concentration (i.e., the condition is set to be the dose at each time of administration, and then the drug is cleared at whatever rate is then defined by the model). If your approach works, that’s obviously fine, but just for informational purposes for the future work.
Author Response
Answers to reviewer 2
We would like to thank review 2 for these comments and suggestions which have led to an improved manuscript, careful review, and for making sure we produce the best academic manuscript we can. We altered the Introduction and Conclusion sections according to the reviewer’s suggestions and completely changed the Abstract. Specifically, we introduce some corrections to Figures 1, 3, 4, add 5. In addition, we added a section (3.5) that outlines in detail the experiment design and model fitting processes. Please note, that all the changes we made can be found in the text highlighted in yellow.
Reviewer 2:
The question that the authors are trying to address is not very clear – “write a model and fit it o data and run simulations” is the “how” of the work, not the “why”. For that, the introduction needs to be worked significantly to clarify the question that can be addressed with these methods (“cure cancer” is not it), gaps in knowledge, how the work fills these gaps, what we now know that we didn’t know before this work, and what the next steps are. These points are not clear.
Answer: Thank you very much for pointing our attention to this shortcoming. We alter the Introduction section according to the suggestions presented in the comment.
- Diagram in Figure 1 seems out of order – as of right now it looks like A, B and C are done separately, and D and E are done separately, and then they converge to F. My understanding of the flow is that A->B->C->D->E->F, and if that is the case, the arrows need to be redrawn.
Answer: We apologize for the lack of clarity in the figure’s design. Thank you for addressing this deficiency. Following this comment, we modified Fig. 1.
- Figure 3: I’m assuming it is reporting the final TGI at the time of when the animals were sacrificed. This is not clear from this diagram. Please redraw it as longitudinal graph, with time on the x axis, and tumor volumes under different regimens on the y-axis. If the graph gets cluttered because of multiple data points, use different subplots.
Answer: Again, thank you for your suggestion – it was really helpful. We alter Fig. 3 according to this comment.
4) Model: the terms are not well justified.
- I take issue with using a mass action term to describe tumor-immune interactions, since it has the obvious flaw of predicting high tumor cell kill even when population sizes of either immune or cancer cells are very small. This is obviously incorrect. I would suggest using a ratio-dependent term, such as described in classic works by DePillis and Radunskaya (xy/(x+y), with appropriate parameters), as it corrects this issue.
Answer: The modification the reviewer suggested is applicable when the population size is approaching zero. Let us explain... In such a case change in the population size (i.e., the system’s parameter) is suffering from the gradient-vanishing phenomenon, which makes the numerical computation less stable and the model less accurate in general. Indeed, in one of our previous studies (S. Bunimovich-Mendrazitsky, S. Halachmi, N. Kronik, Improving bacillus Calmette-guérin (BCG) immunotherapy for bladder cancer by adding interleukin 2 (il-2): A mathematical model, Math. Med. Biol. (2015), 159–188.) Tu was multiplied by gT /(Tu + gT ), which denotes the inversely proportional reduction in killing rate, such that when Tu = 0 the term is equal to 1, and when limTu→∞ gT /(Tu + gT ) = 0. Nevertheless, in the proposed model, the only cell population size reduced to zero is the cancer one which is not present in the discussed term. Thus, in the current study, we decided to use the Michaelis-Menten constant. However, following this comment, in the model design section, we address this choice and provide motivation for our decision.
- The tumor-dependent growth and death terms in the equation for effector cells are not justified – what are the mechanisms this is meant to capture? If you have references, please cite them. If this is purely phenomenological, that’s ok (albeit weaker of course), but please state that clearly.
Answer: Thank you for pointing our attention to this shortcoming. The proposed study’s objective is to model the cytotoxic effect of Cytarabine and Ibrutinib drugs in A20-bearing mice. As a result, we did not measure immune cells but rather used parameters employed by others in the literature to describe the interaction of immune cells with cancer cells (Rodrigues et al. 2019, Kuznetsov et al. 1994, de Pillis et el. 2006). Nevertheless, following this comment, we add to the discussion sections several sentences addressing the reviewer’s questions. Following this comment, we introduced a detailed explanation of this manner to the Discussion Section, Pages 10-11.
- Do we know that both these drugs can cause immune cell mortality to account for the last term in dE/dt? If so, please provide references.
Answer: Ibrutinib inhibits the mutated BTK enzyme responsible for the transformation of B-cells in CLL [22]. Cytarabine is a topoisomerase inhibitor and may affect non-cancerous cells that are dividing, which may include dividing immune cells [24]. Following this comment, we added these explanations to the Introduction section, Page 2.
- Why is the drug cleared through interactions with cancer cells? Please provide explanation and references.
Answer: It is a wonderful question indeed. We did not intend to infer that the drugs are cleared in this way. Rather, clearance will depend on the pharmacokinetics and pharmacodynamics of each drug. In addition, the interaction of the drug with its molecular target will result in its chemical modification and potential neutralization. depend on their metabolism in plasma and within the cell. In the text: “The elimination half-life of each drug was obtained from the literature (www.drugbank.ca)”.
5) Figure 4. Please plot data on the same graphs as the simulation results. A claim that there is “excellent accordance” between data and results is currently unsupported.
Answer: Thank you for your suggestion. Figure 4 simulates the comparative growth rates of untreated versus treated A20 cells for each treatment for the whole mouse. In contrast, Figure 3 represents the inhibition of A20 cell growth as measured in 200,000 cells in 50 ul of blood. Although it is estimated that the whole-body blood volume of Balb/c mice is approximately 2.5 ml, we did not consider this accurate enough for our purposes. Therefore, we could not directly compare the simulation and experimental data in one graph as per your request. Instead, the new Figure 5 compares the growth inhibition of the A20 cells obtained by experiment and simulation for each therapy. Figure 5 shows the excellent accordance between the two results. Following this comment, this explanation has been added to the results section right before Figure 5.
6) Drug synergy is a very specific term, typically defined through the Blys criteria. I am assuming this was not the criterion used here? If so, please change the wording as it currently is not correct to describe the projected impact of the combination.
Answer: You are right... So we have replaced the term "synergy" with the term "combination therapy" throughout the text, which we find really more appropriate.
7) Mathematical analysis in the appendix is nice but not very helpful – stability analysis is not very informative at this stage. Ok, solution exists, and we know its stability. What does that tell us about the biology? What are the important parameters responsible for where the equilibrium will be? What insights were gained from the stability analysis, or is it just a mathematical formality?
- More informative would be sensitivity analysis (which parameters and corresponding mechanisms are particularly sensitive to changes in system values, to changes in treatment)? Identifiability analysis would be more informative as well – how do you know these parameter combinations are unique?
Answer: We totally agree with you. Unfortunately, in our previous attempts, stability analysis failed to provide any interesting results. Following this comment, we performed a sensitivity analysis for the most sensitive parameter – mu_AC, as one can see in the newly introduced Section 3.5.
8) Parameter estimates: how were they obtained? Methods are missing. Did you use fmincon? Monte Carlo Simulations? Found them by hand?
Answer: Thank you for pointing this shortcoming out. Some of the parameters were taken from previous studies as shown in Table 1 in the manuscript. Several other parameters were derived from the experiments – in particular, the r, d, and the initial conditions for the simulation. Lastly, several parameters have been manually obtained from numerical simulations. Following this comment, we formally explain how we obtained each parameter in the model as shown in Sections 3.3-3.5.
9) There is no analysis of what may be driving “synergy”, or additivity, or anything else. What terms are responsible? What are the possible biological interpretations?
Answer: As mentioned in the answer to comment #6, we replaced this term. In addition, the interaction of different drugs to produce additive biological responses is highly complex and we did not intend to address it in this study. In this work’s scope, we only describe the phenomenon between Cyt and Ibr which has not been previously reported. Following this comment, we clearly stated this shortcoming as a limitation of the proposed work and suggested it as possible future research.
10) Paper is severely undercited, and primarily cites mathematical papers, which would perhaps be appropriate (with the addition of sensitivity and potentially also identifiability analysis) in a mathematical-leaning journal but not in Cells. This model is potentially a good start but it is currently very far from being useful for biological purposes to even be listed in “methods”.
Answer: Thank you for your comment. This study was submitted to Cells specifically because the editors of the special issue were interested in mathematical models for cancer therapy. Nonetheless, we find your comment really important and as a result, we have added a number of biologically related references:
- Kareva, I.; Karev, G. From experiment to theory: what can we learn from growth curves? Bulletin of mathematical biology 2018, 80, 151–174.
- Beckman, R.A.; Kareva, I.; Adler, F.R. How should cancer models be constructed? Cancer Control 2020, 27, 1073274820962008.
- de Pillis, L.G.; Gu, W.; Radunskaya, A.E. Mixed immunotherapy and chemotherapy of tumors: modeling, applications and biological interpretations. Journal of theoretical biology 2006, 238, 841–862.
- Antipov, A.; Bratus, A.S. Mathematical model of optimal chemotherapy strategy with allowance for cell population dynamics in a heterogeneous tumor. Computational mathematics and mathematical physics 2009, 49, 1825–1836.
- Bratus, A.; Todorov, Y.; Yegorov, I.; Yurchenko, D. Solution of the feedback control problem in the mathematical model of leukaemia therapy. Journal of Optimization Theory and Applications 2013, 159, 590–605.
- Kuznetsov, M.; Clairambault, J.; Volpert, V. Improving cancer treatments via dynamical biophysical models. Physics of Life Reviews 2021, 39, 1–48.
- Berezansky, L.; Bunimovich-Mendrazitsky, S.; Domoshnitsky, A. A mathematical model with time-varying delays in the combined treatment of chronic myeloid leukemia. Advanced in Difference Equations 2012, 1, 1–13.
- Agur, Z.; Elishmereni, M.; Kheifetz, Y. Personalizing oncology treatments by predicting drug efficacy, side-effects, and improved therapy: mathematics, statistics, and their integration. Wiley Interdisciplinary Reviews: Systems Biology and Medicine 2014, 6, 239–253.
- Agur, Z.; Halevi-Tobias, K.; Kogan, Y.; Shlagman, O. Employing dynamical computational models for personalizing cancer immunotherapy. Expert opinion on biological therapy 2016, 16, 1373–1385.
- Rabian, F.; Lengline, E.; Rea, D. Towards a personalized treatment of patients with chronic myeloid leukemia. Current Hematologic Malignancy Reports 2019, 14, 492–500.
- Foss, B.; Ulvestad, E.; Hervig, T.; Bruserud, O. Effects of cytarabine and various anthracyclins on platelet activation: characterization of in vitro effects and their possible clinical relevance in acute myelogenous leukemia. International journal of cancer 2002, 97, 106–114.
- Aslan, B.; Kismali, G.; Chen, L.S.; Iles, L.R.; Mahendra, M.; Peoples, M.; Gagea, M.; Fowlkes, N.W.; Zheng, X.;Wang, J.; others. Development and characterization of prototypes for in vitro and in vivo mouse models of ibrutinib-resistant CLL. Blood advances 2021, 5, 3134–3146.
- Kuznetsov, V.A.; Makalkin, I.A.; Taylor, M.A.; Perelson, A.S. Nonlinear dynamics of immunogenic tumors: parameter estimation and global bifurcation analysis. Bulletin of mathematical biology 1994, 56, 295–321.
11) Technical aspect: typically the easiest way to implement modeling drug administration is through change in initial conditions for the equation for drug concentration (i.e., the condition is set to be the dose at each time of administration, and then the drug is cleared at whatever rate is then defined by the model). If your approach works, that’s obviously fine, but just for informational purposes for the future work.
Answer: Thank you very much for this suggestion. We were not aware of this approach and find it fascinating. We will take this into account in our future studies.

Reviewer 3 Report
Guev & coll describes the in-mouse kinetics of A20 mCherry cells after treatment with cytarabine or ibrutinib. the previously validated differential equation model developed by the authors for in-vitro drug effect allowed to predict also in-vivo kinetics and to predict synergy between cytarabine and ibrutinib.
The presented methods are interesting for exploring new drug associations, however cytarabine is not considered a partner for ibrutinib due to its toxicity. Therefore, the discussion section should be greately expanded in order to consider the potential clinical applications of the study results.
Round 2
Reviewer 1 Report
In the reference omit publications on the same investigations. 1.E.K. Afenya, C.P. Calderon (Comm. Comm. Theor.Biol. 8 (2), 2003, 225-233.
2. A.S. Bratus and others. Nonlinear Analysis: Real World Application 13, 2012,1044-1039.
3. Y Todorov and others. Russian Journal of Numerical Anal. and Math . Mod. 2012, v.26, #6, 599-604.
4. E. Fimmel and others. Math. Biosc. Engin. 2013, 10-1,151-165.
Reviewer 2 Report
.
Author Response
The attachment is the reply to R1 AND R3.
